# Impact of *Ginkgo biloba* drug interactions on bleeding risk and coagulation profiles: A comprehensive analysis

Ngo Thi Quynh Mai[1], Nguyen Viet Hieu[1], Tran Thi Ngan [1,2], Tran Van Anh[1,2], Pham Van Linh[3], Nguyen Thi Thu Phuong [1,2]*

1 Faculty of Pharmacy, Hai Phong University of Medicine and Pharmacy, Hai Phong, Vietnam,
2 Department of Pharmacy, Hai Phong International Hospital, Hai Phong, Vietnam, 3 Department of Pathology and Immunology, Hai Phong University of Medicine and Pharmacy, Hai Phong, Vietnam

* nttphuong@hpmu.edu.vn

## Abstract

This retrospective observational study was conducted to investigate the prevalence and clinical implications of drug interactions involving *Ginkgo biloba* extract on bleeding risk and coagulation profiles. Our analysis utilized data from patients admitted to Hai Phong International Hospital between January 2022 and December 2023. Inclusion criteria consisted of patients aged 18 years and above, those prescribed *Ginkgo biloba* extract alone or in combination with other medications, and the availability of complete medical records, including medication history, laboratory tests, and clinical outcomes. Out of 2,647 prescriptions meeting the inclusion criteria, 342 exhibited drug interactions with a prevalence rate of 12.94%. Notably, *Ginkgo biloba* extract frequent interacts with antiplatelets, anticoagulants, and nonsteroidal anti-inflammatory drugs, with Clopidogrel and Aspirin exhibiting the highest prevalence rates of 2.61% each. However, interactions with anticoagulants including direct oral anticoagulants and acenocoumarol, were not statistically significant in our analysis. Omeprazole was a frequently interacting drug (2.34%) of mild severity. Among the 747 patients analyzed for bleeding disorders, 31 (4.15%) exhibited bleeding symptoms. Correlation analysis indicated a strong association between clinical bleeding and abnormal coagulation results (OR, 1.75; p < 0.001). Moreover, significant correlations were found between *Ginkgo biloba* extract drug interactions and the bleeding risk (OR: 1.08, p < 0.001) and abnormal coagulation (OR: 1.49, p < 0.001). The severity of *Ginkgo biloba* extract drug interactions did not correlate with bleeding risk (OR: 1.01, p = 0.767) but influenced abnormal coagulation test results (OR: 0.813, p = 0.019). Specific medications, including clopidogrel, aspirin, celecoxib, loxoprofen, nifedipine, and omeprazole, were significantly associated with the risk of bleeding and abnormal coagulation (p < 0.05). Interactions with ticagrelor, etoricoxib, insulin, omeprazole, and domperidone were associated with abnormal coagulation tests without affecting the reported bleeding. These findings underscore the critical need of evaluating potential interactions involving *Ginkgo biloba* extract drug interactions in clinical pratice, particularly when assessing bleeding risk and managing coagulation.

**Data availability statement:** All relevant data are within the manuscript and its Supporting Information files.

**Funding:** The author(s) received no specific funding for this work.

**Competing interests:** The authors have declared that no competing interests exist.

## Introduction

*Ginkgo biloba* extract, a supplement from *Ginkgo biloba* (*G. biloba*) leaves, has gained popularity worldwide owing to its perceived health benefits, including cognitive enhancement and peripheral circulation improvement. *G. biloba* leaves contain two primary active ingredients: terpene lactones (including ginkgolides and diterpenes) and Ginkgo flavone glycosides (containing ginkgetin, bilobetin, and sciadopitysin), which are present in varying concentrations [1]. *G. biloba* leaf extract (EGb) is widely recognized as one of the most popular health supplements globally due to its benefits for mental focus. EGb 761, the standardized EGb, is commonly used in studies investigating its effects. Research has indicated that EGb influences several neurotransmitter pathways and brain structures, particularly those observed in animal studies [2,3]. EGb 761 reduces stress-induced corticosterone hypersecretion in rats by decreasing the number of peripheral adrenal benzodiazepine receptors [4]. However, the efficacy of EGb 761 in treating dementia has been a topic of debate, with conflicting findings from various studies. A 52-week randomized, double-blind, placebo-controlled trial involving 309 patients suggested that EGb 761 was safe and, albeit modestly, appeared to stabilize and improve cognitive performance and social functioning in patients with dementia for 6 months to 1 year [5]. Systematic reviews conducted in 2009 and 2018, comprising 36 and 38 trials, respectively, highlighted the safety of EGb but did not find sufficient evidence to support its clinical benefit in patients with cognitive impairment and dementia [6]. Conversely, a 2015 systematic review of nine trials suggested that EGb 761 at a dose of 240 mg/day could slow the decline in cognition, function, behavior, and overall global change at 22–26 weeks, particularly in patients with neuropsychiatric symptoms [7]. Overall, EGb is regarded as safe and generally well-tolerated, with a maximum recommended dose of 240 mg/day [8]. Mild side effects may include headache, heart palpitations, gastrointestinal issues, constipation, and allergic skin reactions [1]. Although a systematic review and meta-analysis found no significant effect of EGb on prothrombin time, activated partial thromboplastin time, or platelet aggregation, several case reports have noted a temporal link between EGb use and bleeding events, including severe intracranial bleeding [9,10].

Although is commonly used, it carries certain risks, particularly regarding its potential interactions with prescription medications. Such interactions may result in complications like bleeding disorders and abnormal coagulation profiles [11]. Therefore, it is important to exercise caution when administering EGb to patients with bleeding disorders or those using NSAIDs, antiplatelet drugs, or anticoagulants [12,13].

Recently, several case reports have indicated a possible association between hemorrhagic complications and the use of EGb preparations. In light of these concerns, a trial was conducted to assess the effects of the *G. biloba* special extract, EGb 761, on hemostatic parameters. The results revealed that none of the 29 coagulation and bleeding parameters evaluated showed any evidence of EGb 761 inhibiting blood coagulation or platelet aggregation. Furthermore, the study did not reveal any evidence to substantiate the causal relationship between EGb administration of EGb 761 and hemorrhagic complications [14]. However, the concomitant use of EGb with prescription drugs has raised concerns among healthcare professionals owing to the potential for pharmacokinetic and pharmacodynamic interactions. These interactions can alter the efficacy and safety profiles of EGb and co-administered medications, posing significant challenges in clinical management. Concerning perioperative use, there is insufficient evidence regarding the risks associated with EGb use. However, the study suggested that physicians should consider discontinuing EGb for at least 36 h before a planned surgical procedure [11]. However, the concomitant use of EGb with prescription drugs has

raised concerns among healthcare professionals owing to the potential for pharmacokinetic and pharmacodynamic interactions. These interactions can alter the efficacy and safety profiles of EGb and co-administered medications, posing significant challenges in clinical management.

Through a retrospective observational study design, we aimed to elucidate the frequency and severity of EGb drug interactions, with a particular focus on their association with bleeding risk and abnormalities in coagulation profiles in a real-world clinical setting. Ultimately, the findings of this study have the potential to guide clinical decision-making, optimize patient care, and enhance medication safety in individuals receiving EGb therapy concurrently with prescription medications.

## Materials and methods

### Study design

This retrospective observational study analyzed data collected from the medical records of patients admitted to Hai Phong International Hospital (Hai Phong, Vietnam) between January 2022 and December 2023. Data was accessed for study purposes on February 15, 2024.

**Inclusion criteria.**

- Patients aged 18 years and above.

- Patients prescribed EGb alone or in combination with other prescription drugs.

- Availability of complete medical records, including medication history, laboratory tests, and clinical outcomes. This criterion was only applied to investigate the effect of EGb interactions on the occurrence of bleeding or abnormal coagulation tests.

**Exclusion criteria.**

- Patients with incomplete medical records.

- Patients with a history of bleeding disorders unrelated to medication use.

- Patients prescribed herbal supplements other than EGb.

- Patients with missing demographic or clinical data.

**Data extraction.** A structured data extraction form was utilized to collect the following information:

- Patient demographics (age, gender, weight, body mass index [BMI])

- Medication history, including EGb and other prescription drugs.

- Laboratory tests, including activated partial thromboplastin time (APTT), prothrombin time (PT), and fibrinogen levels

- Clinical outcomes such as bleeding events and diagnosis of bleeding disorders

- Severity classification of drug interactions based on established criteria.

**Identification of drug interactions.** Drug interactions involving EGb were identified using databases such as UpToDate, Micromedex, and Drugs.com. Interactions were classified based on severity (mild, moderate, or severe) and documented adverse effects, particularly bleeding disorders, and abnormal coagulation profiles.

## Statistical analysis

Descriptive statistics were used to summarize the patient demographics, medication profiles, and clinical outcomes. The prevalence of drug interactions with EGb was calculated as the proportion of patients with documented interactions. Chi-square tests or Fisher's exact tests were used to assess associations between categorical variables, whereas t-tests or Mann–Whitney U tests were used for continuous variables. Logistic regression analysis was performed to identify the factors associated with bleeding events and abnormal coagulation test results. Statistical significance was set at $p < 0.05$. The logistic regression model was built using the R statistical software, version 3.2.4 (A Language and Environment for Statistical Computing, Vienna, Austria) [15].

## Ethical considerations

This study adhered to the ethical principles outlined in the Declaration of Helsinki. Patient confidentiality and privacy were maintained throughout the data collection and analysis. Institutional Review Board (IRB.23.128) approval was obtained before the commencement of data collection and analysis. All patients provided verbal consent for the use of their data in research at the time of their initial treatment. This consent included an explanation of the potential use of their anonymized data in future research studies.

## Results

To calculate the frequency and rate of drug interactions between EGb and other prescription drugs, we selected 2,647 prescriptions that met the inclusion and exclusion criteria. Among these, 342 prescriptions exhibited drug interactions, resulting in a prevalence of 12.94%. The study diagram is presented in Fig 1.

The interaction between EGb and conventional drugs are presented in Table 1. EGb interacted predominantly with antiplatelet drugs, anticoagulants, and NSAIDs. Interactions with antiplatelet drugs manifested the highest prevalence rate, coupled with an average severity level. Specifically, interactions with clopidogrel and aspirin were documented in 69 prescriptions, representing a rate of 2.61%.

Anticoagulants also demonstrated notable interaction prevalence, particularly with direct oral anticoagulants (DOACs), documented in 20 prescriptions (18%), and with acenocoumarol, noted in 13 prescriptions (0.49%). According to UpToDate and Micromedex, interactions between EGb and agents with antiplatelet properties or anticoagulants may enhance the risk of adverse or toxic effects, such as bleeding.

Within the spectrum of interactions between EGb and other drugs, omeprazole was identified with notable frequency, appearing in 62 prescriptions (2.34%), though its severity was classified as mild.

Of the 2,647 patients prescribed EGb preparations, 747 met the criteria for participation in Part 2 of the study aimed at exploring the relationship between drug interactions involving EGb and bleeding disorders. Of these patients, 489 individuals without drug interactions were assigned to the control group, while the group with drug interactions comprised 258 patients. Table 2 presents the characteristics of these patients, including weight, age, and sex, categorized into two groups: those with drug interactions involving EGb and other drugs and those without such interactions. Of these, 31 (4.15%) were diagnosed with bleeding disorders or related symptoms. For this subset of patients, the following mean (±SD) coagulation test values were observed: APTT: 57.25 ± 9.34 seconds, PT (%): 110.71 ± 26.18, and fibrinogen level: 3.56 ± 1.25 g/L.

To further clarify the relationship between drug interactions involving EGb and bleeding disorders, a control group of 484 patients without drug interactions was selected. In

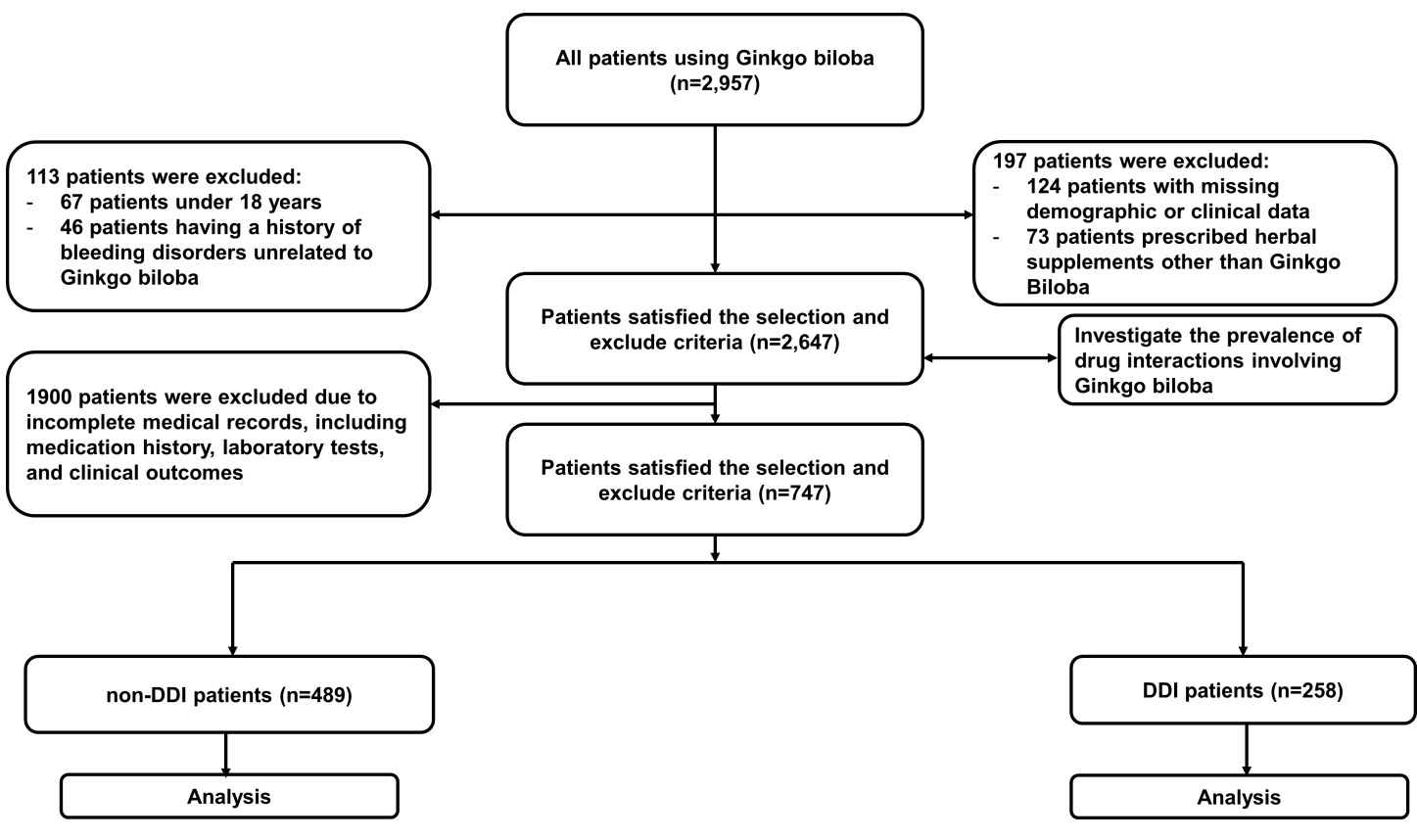

**Fig 1. Study diagram.**

comparison, 258 patients exhibited drug interactions between EGb and other drugs. The distribution of interaction pairs was as follows: 185 patients (61.24%) had one interaction pair, 64 patients (24.81%) had two interaction pairs, 7 patients (2.71%) had three interaction pairs, and 2 patients (0.77%) had four interaction pairs. Of the 258 patients experiencing drug-drug interactions involving EGb, 226 (87.6%) individuals were identified with moderate severity interactions, while only 32 patients (12.4%) exhibited mild severity interactions (Table 2).

We conducted a correlation analysis to examine the relationship between clinical variables and drug interaction characteristics and the occurrence of clinically documented bleeding and abnormalities in coagulation test results among 747 patients. The findings, summarized in Table 3, indicate that weight and BMI did not significantly influence the occurrence of clinically recorded bleeding or abnormalities in coagulation tests (p > 0.05). Although sex did not demonstrate a substantial association with the occurrence of bleeding, it exhibited a slight correlation with abnormalities in coagulation test results, with an odds ratio (OR) of 1.07 (95% CI: 0.99; 1.15) (p = 0.071).

A significant association was observed between cases of clinical bleeding and abnormal coagulation results, with an OR of 1.75 (95% CI: 1.48; 2.08), indicating a strong correlation (p < 0.001). Each coagulation test, including APTT, PT, and fibrinogen level, demonstrated a close relationship with the manifestation of clinical bleeding.

Furthermore, a significant link between drug interactions involving EGb and prescription medications and the occurrence of bleeding and abnormal coagulation test results. The odds ratios for these associations were found to be 1.08 (95% CI: 1.04; 1.11) and 1.49 (95% CI: 1.38;

**Table 1. Interaction between EGb and conventional drugs (n = 2,647).**

| Drug class | Prescribed drug | Number of DDIs | Frequency (%) | Severity | Mechanism | Consequence | Reference |
|---|---|---|---|---|---|---|---|
| Agents with antiplatelet properties | Clopidogrel | 69 | 2.61 | Moderate | Many herbal products have been shown to inhibit platelet function, prolong bleeding time, or contribute to bleeding events in people. Concomitant use of these herbal products with drugs that have a similar pharmacologic potential may increase the risk of bleeding. | Enhance the adverse/toxic effects of agents with antiplatelet properties. Bleeding may occur | UpToDate |
| | Aspirin | 69 | 2.61 | Moderate | | | |
| | Ticagrelor | 2 | 0.08 | Moderate | | | |
| Anticoagulants | DOAC (enoxaban/ rivaroxaban) | 18 | 0.68 | Moderate | Many herbal products possess the ability to cause or potentiate bleeding (inhibit clotting/coagulation or primary hemostasis) by one of several mechanisms. They may inhibit platelet aggregation, inhibit cyclooxygenase activity, interfere with one or more components of the coagulation cascade, or increase bleeding risk by another mechanism. The concomitant use of such herbs with other herbs or drugs possessing a similar pharmacologic potential may increase the risk of bleeding. | Enhance the adverse/toxic effects of anticoagulants. Bleeding may occur. | UpToDate |
| | Acenocoumarol | 13 | 0.49 | Moderate | | | |
| Nonsteroidal anti-inflammatory drugs (NSAIDs) | Etoricoxib | 20 | 0.76 | Moderate | Many herbal products have been shown to inhibit platelet function, prolong bleeding time, or contribute to bleeding events in people. Concomitant use of these herbal products with drugs that have a similar pharmacologic potential (such as NSAIDs) may increase the risk of bleeding. | Enhance the adverse/toxic effects of NSAIDs. Bleeding may occur | UpToDate |
| | Celecoxib | 14 | 0.53 | Moderate | | | |
| | Meloxicam | 10 | 0.38 | Moderate | | | |
| | Loxoprofen | 2 | 0.08 | Moderate | | | |
| Other | Nifedipine | 24 | 0.91 | Moderate | Inhibition of cytochrome P450 3A4 by *G. biloba* | Concurrent use of *G. biloba* and nifedipine may result in an increased risk of nifedipine side effects. | UpToDate |
| | Insulin | 22 | 0.83 | Moderate | *G. biloba* may increase pancreatic beta-cell function | Concurrent use of *G. biloba* and insulin may result in altered insulin effectiveness. | Micromedex |
| | Omeprazole | 62 | 2.34 | Minor | The mechanism of this potential interaction is unknown. While *G. biloba*-mediated induction of CYP2C19 has been proposed as a potential mechanism, the large effect seen in CYP2C19 poor metabolizers combined with a lack of effects seen with voriconazole and diazepam (substrates of CYP2C19) in other studies [3,4] makes the role of CYP2C19 induction in this interaction questionable. Additionally, because natural products have variable constituents, it is unknown if a particular formulation or dose is more susceptible to this interaction. | Decrease the serum concentration of omeprazole | Micromedex |
| | Domperidone | 17 | 0.64 | Minor | Inhibition of CYP3A4-mediated domperidone metabolism | Concurrent use of domperidone and *G. biloba* may result in increased domperidone exposure and an increased risk of QT prolongation. | Micromedex |
| Total | | 342 | 12.94 | | | | |

DDIs: Drug-drug interactions; DOAC: Direct oral anticoagulant; NSAIDs: Nonsteroidal anti-inflammatory drugs.

1.58), respectively, implying a significant correlation (p < 0.001). These findings underscore the clinical relevance of drug interactions, particularly those involving EGb, in the context of bleeding risk and abnormal coagulation profiles.

**Table 2. Clinical characteristics and drug interactions of the patient group to evaluate the relationship between DDI of EGb - bleeding/coagulopathy (n = 747).**

|  | No DDI (n = 489) | DDI (n = 258) | Total (n = 747) |
|---|---|---|---|
| Weight (kg, mean±SD) | 57.14 ± 9.44 | 57.46 ± 9.15 | 57.25 ± 9.34 |
| Age (years, mean±SD) | 54.26 ± 15.79 | 63.08 ± 13.93 | 57.31 ± 15.74 |
| Sex (Ref: female) (n,%) | 331 (67.69%) | 172 (66.67%) | 503 (67.34%) |
| BMI (mean±SD) | 22.62 ± 2.81 | 23.09 ± 2.78 | 22.78 ± 2.81 |
| Symptom of bleeding (Ref: yes) (n, %) | 8(1.64%) | 23(8.91%) | 31(4.15%) |
| APTT (s) (mean±SD) | 33.3 ± 4.56 | 34.11 ± 5.45 | 33.78 ± 5.12 |
| PT (%, mean±SD) | 102.2 ± 17.31 | 126.83 ± 31.92 | 110.71 ± 26.18 |
| Fibrinogen (g/l, mean±SD) | 3.19 ± 0.7 | 5.24 ± 1.78 | 3.55 ± 1.25 |
| Abnormal coagulant test (n, %) | 127 (25.97%) | 169 (65.5%) | 296 (39.63%) |
| Number of DDIs (n, %) |  |  |  |
| One interaction pair |  | 185 (71.71%) | 185 (24.77%) |
| Two interaction pairs |  | 64 (24.81%) | 64 (8.57%) |
| Three interaction pairs |  | 7 (2.71%) | 7 (0.94%) |
| Four interaction pairs |  | 2 (0.78%) | 2 (0.27%) |
| Severity of DDI (n = 258) |  |  |  |
| Moderate |  | 32 (12.4%) | 32 (4.28%) |
| Minor |  | 226 (87.6%) | 715 (95.72%) |

DDIs: Drug-Drug interactions; SD: Standard deviation; BMI: Body mass index; APTT: Activated partial thromboplastin time; PT: Prothrombin time.

In our study, we observed no correlation between the severity of drug interactions involving EGb and prescription drugs and the occurrence of bleeding, with an OR of 1.01 (95% CI: 0.94; 1.08), indicating no significant association (p = 0.767). However, this variable demonstrated a statistically significant effect on abnormalities in coagulation test results, with an OR of 0.813 (95% CI: 0.68; 0.96) and a p-value of 0.019. This suggests that although the severity of drug interactions may not influence the risk of bleeding, it does have a notable impact on the abnormalities observed in the coagulation test results.

When examining the relationship between drug interactions involving EGb and specific medications and the occurrence of bleeding, our analysis revealed that several drugs exhibited statistical significance with a p-value of less than 0.05. These drugs include clopidogrel, aspirin, celecoxib, loxoprofen, nifedipine, and omeprazole. Significant drug interactions involving EGb were observed with clopidogrel (OR: 1.10, p < 0.001), aspirin (OR: 1.12, p < 0.001), and omeprazole (OR: 1.10, p = 0.0003). Additionally, interactions of EGb with ticagrelor, etoricoxib, insulin, omeprazole, and domperidone were observed to be associated with abnormalities in coagulation tests without affecting physician-reported bleeding.

## Discussion

Our findings provide valuable insights into the prevalence, severity, and clinical implications of drug interactions involving EGb and prescription medications. We observed that 12.94% of the prescriptions exhibiting interactions with EGb, highlighting the need for increased awareness among clinicians. A retrospective analysis utilizing data from the Taiwan National Health Insurance Research Database spanning from 2000 to 2008 identified a gradual increase in the concurrent use of EGb with antiplatelet or anticoagulant agents [16]. These trends reflect the growing popularity of EGb and underscore the necessity of monitoring potential herb-drug interactions in clinical practice.

**Table 3. Association of EGb drug interactions with bleeding and abnormal coagulation tests (n = 747).**

| Variables | The occurrence of bleeding | | Abnormal coagulant test | |
|---|---|---|---|---|
| | OR (CI 5; 95%) | P-value | OR (CI 5; 95%) | P-value |
| Weight | 1 (0.99; 1) | 0.607 | 1 (0.99; 1) | 0.12 |
| Sex (Ref: Male) | 1 (0.97; 1.03) | 0.733 | 1.07 (0.99; 1.15) | 0.071 |
| Age (years) | 1 (0.99; 1) | 0.325 | 10024 (1; 1004) | 0.029 |
| BMI (mean±SD) | 1 (0.99; 1) | 0.285 | 1 (0.99; 1.02) | 0.198 |
| Occurrence of bleeding (Ref: yes) | | | 1.75 (1.48; 2.08) | <0.001 |
| APTT (s) | 1 (0.99; 1) | 0.084 | 1.03 (1.03; 1.04) | <0.001 |
| PT (%) | 1.0009 (1.0004; 1.001) | 0.0005 | 1005 (1004; 1007) | <0.001 |
| Fibrinogen (g/l) | 1.04 (1.02; 1.05) | <0.001 | 1.22 (1190; 1.25) | <0.001 |
| Abnormal coagulant test (Ref: yes) | 1.09 (1.06; 1.12) | <0.001 | | |
| DDI (Ref: Yes) | 1.08 (1.04; 1.11) | <0.001 | 1.49 (1.38; 1.58) | <0.001 |
| Number of DDIs | 1.06 (1.04; 1.09) | <0.001 | 1.36 (1.3; 1.42) | <0.001 |
| Severity of DDIs (Ref: Moderate) | 1.01 (0.94; 1.08) | 0.767 | 0.813 (0.68; 0.96) | 0.019 |
| Clopidogrel | 1.1 (1.05; 1.15) | <0.001 | 1.58 (1.4; 1.77) | <0.001 |
| Aspirin | 1.12 (1.07; 1.18) | <0.001 | 1.48 (132; 1.66) | <0.001 |
| Ticagrelor | 0.96 (0.73; 1.27) | 0.769 | 1.83 (0.92; 3.61) | 0.081 |
| DOAC (enoxaban/rivaroxaban) | 1.01 (0.92; 1.11) | 0.763 | 1.11 (0.88; 1.39) | 0.363 |
| Acenocoumarol | 0.96 (0.86; 1.07) | 0.45 | 1.07 (0.81; 1.39) | 0.628 |
| Etoricoxib | 1 (0.92; 1.1) | 0.847 | 1.37 (1.1; 1.69) | 0.0048 |
| Celecoxib | 1.11 (0.99; 1.23) | 0.055 | 0.48 (1.15; 0.92) | 0.0026 |
| Meloxicam | 1.06 (0.94; 1.2) | 0.351 | 1.23 (0.91; 1.67) | 0.185 |
| Loxoprofen | 1.58 (1.2; 2.08) | 0.00111 | 1.83 (0.93; 3.61) | 0.0807 |
| Nifedipine | 1.09 (1; 1.18) | 0.0371 | 1.16 (0.95; 1.42) | 0.139 |
| Insulin | 1 (0.92; 1.09) | 0.925 | 1.34 (1.09; 1.65) | 0.0054 |
| Omeprazole | 1.10 (1.04; 1.15) | 0.0003 | 1.7 (1.51; 1.92) | <0.001 |
| Domperidone | 1.01 (0.92; 1.12) | 0.717 | 1.29 (1.02; 1.63) | 0.0325 |

OR: Odds Ratio; CI: Confidence interval; BMI: Body mass index; SD: Standard deviation; APTT: Activated partial thromboplastin time; PT: Prothrombin time; DDIs: Drug-Drug interactions.

Among the 195 consecutive patients surveyed in this study, EGb was identified as the most frequently used herbal medicine. Of the identified herb-drug interactions, eight cases involved interactions between EGb and aspirin (acetylsalicylic acid), while one case was linked to trazodone [17]. In line with our findings, these interactions were mainly observed with antiplatelet drugs, anticoagulants, and NSAIDs. Such findings reinforce the need for clinicians to recognize and mitigate potential EGb-drug interactions in practice.

The majority of EGb-related drug interactions in this study were classified as moderate (87.6%), with 12.4% categorized as mild, such as interactions with omeprazole. Moderate interactions, particularly those involving anticoagulants, pose significant risks, as they can increase bleeding potential even without immediate clinical symptoms [18,19]. Mild interactions, like those with omeprazole, while less severe, may affect drug efficacy through pharmacokinetic mechanisms such as CYP2C19 modulation. These findings highlight the importance of vigilance in monitoring both moderate and mild interactions, as even seemingly minor effects can compromise patient safety.

Interactions with antiplatelet drugs, such as clopidogrel and aspirin, were particularly notable in our study, accounting for 69 prescriptions (2.61%) and association with adverse outcomes, including bleeding events and coagulation abnormalities. These interactions were

the most frequent due to the widespread use of clopidogrel and aspirin in managing cardio-vascular diseases, particularly in patients requiring dual antiplatelet therapy [20]. Additionally, aspirin's well-documented role in increasing bleeding risks may be exacerbated when com-bined with EGb, given its known anticoagulant and antiplatelet properties [21]. Our findings are consistent with prior reports documenting severe bleeding events, including intracranial hemorrhages, associated with EGb use [22,23]. However, controlled trials investigating the anticoagulant effects of Ginkgo have produced inconsistent findings. For instance, Stanger et al. (2012) reported no significant enhancement of antiplatelet activity when *G.biloba* was com-bined with clopidogrel or cilostazol, though it did potentiate cilostazol's effect on prolonging bleeding time [24,25].

In our study, no correlation was observed between age and bleeding risk, which contrasts with previous findings from Chan et al., where patients aged ≥ 65 and male patients using *G.biloba* exhibited higher odds of hemorrhage (adjusted OR: 3.8 and 1.4, respectively). This discrepancy may stem from the younger average age of 60 years in our cohort, which could have minimized age-related differences [16]. Such homogeneity may have minimized age-related variations in bleeding risk typically observed in older populations.

The frequent co-administration of EGb and aspirin, both commonly used by older adults for cognitive enhancement and cardiovascular disease prevention, raises critical safety con-cerns. Given the increased bleeding risks associated with their combined use, as documented in both case reports [22] and observational studies [26], clinicians should carefully evaluate this combination, particularly in populations with higher baseline bleeding risks. Further underscoring the risks, a cross-sectional survey demonstrated that 21% of patients co-ingested herbs such as EGb with antiplatelet or anticoagulant therapies, with nearly half at risk of sig-nificant drug-herb interactions [27].

Our findings also indicate that EGb interactions with anticoagulants, such as DOACs, are noteworthy, though no significant correlation was observed with acenocoumarol. This finding aligns with previous in vivo studies, which have not consistently demonstrated significant interactions between G. biloba and CYP2C9 substrates, including warfarin [28]. However, case reports and observational studies have suggested a potential increased bleeding risk with concurrent use of EGb and warfarin [29,30]. While Stoddard et al. reported a significant increase in bleeding events (hazard ratio = 1.38, 95% CI: 1.20 to 1.58, p < 0.001) with this combination, the absolute increase in bleeding events was relatively small [13]. These mixed results emphasize the need for further studies.

Notably, while the interaction between EGb omeprazole was, its severity were classified as mild. Previous studies have suggested that EGb induces omeprazole hydroxylation through CYP2C19 modulation, potentially reducing the efficacy of omeprazole or other CYP2C19 sub-strates [31]. Although not immediately clinically significant, this interaction warrants further investigation, particularly in patients with genetic polymorphisms affecting CYP2C19 activity.

Our analysis revealed that the severity of drug interactions did not correlate with bleeding risk but significantly impacted coagulation test abnormalities. While some studies suggest that *G. biloba* may decrease markers of intravascular coagulation [32] and enhance anti-platelet effects when combined with other agents [33], others report no significant impact on coagulation parameters or bleeding risk, such as in a randomized controlled trial involving Alzheimer's patients [34]. Despite these findings, 15 case reports have described a temporal correlation between *G. biloba* administration and bleeding events including eight cases of intracranial hemorrhage [10]. Nevertheless, other risk factors for bleeding were identified among the 13 case reports. Six reports documented the cessation of *G. biloba* intake, which resulted in the absence of recurrent bleeding episodes. Additionally, three reports indicated elevated bleeding times in patients with *G. biloba* intake [10]. This finding underscores the

importance of monitoring coagulation parameters in patients with EGb-related drug interactions, even when no clinical bleeding is evident.

Our findings emphasize the need for clinicians to assess potential EGb interactions carefully, particularly when prescribing to patients on antiplatelet or anticoagulant therapies. Regular monitoring of coagulation parameters, such as APTT, PT, and fibrinogen levels, is essential for high-risk patients to detect and manage coagulation abnormalities early. Additionally, clinicians should educate patients about the risks of self-medicating with EGb and emphasize the importance of disclosing all herbal supplement use. A cautious approach to prescribing EGb in populations with elevated bleeding risks is imperative, and its use should be avoided unless clinically justified and closely supervised. These proactive measures are vital for minimizing adverse outcomes and ensuring patient safety.

The study has notable strengths, including its large sample size of 2,647 prescriptions, which provides a solid foundation for robust statistical analyses. By focusing on real-world clinical data, the study offers insights that are directly applicable to everyday medical practice. The detailed evaluation of both the frequency and severity of drug interactions allows for a deeper understanding of EGb's clinical impact. Moreover, the inclusion of interactions beyond bleeding risks broadens the scope of the findings, highlighting the complexity of EGb-related interactions. Despite its strengths, the study has several limitations. First, the study sample was derived from patients who were prescribed EGb; this may have introduced selection bias. Patients who choose to use or are prescribed EGb may possess different characteristics compared to those who do not use the supplement, potentially impacting the generalizability of our findings. Second, incomplete laboratory data, particularly for APTT, PT, and fibrinogen levels, could affect the accuracy of our results. Additionally, uncontrolled confounding variables such as comorbidities, concurrent medications, lifestyle habits, and dietary patterns, which may have influenced the associations observed in our study, were not thoroughly assessed.

Future studies should focus on confirming EGb effects on coagulation and bleeding risks through large-scale randomized trials. Research should explore its mechanisms, particularly its impact on platelet function and CYP450 enzyme modulation, and the role of genetic polymorphisms in interaction susceptibility. Developing evidence-based guidelines for safe EGb use, especially in high-risk populations such as older adults or those on anticoagulants, is critical to improving clinical outcomes.

## Conclusion

In conclusion, this retrospective observational study provides valuable insights into the prevalence and implications of drug interactions involving EGb on bleeding risk and coagulation profiles. We found a notable prevalence of drug interactions, primarily involving antiplatelets, anticoagulants, and NSAIDs. Interestingly, interactions with anticoagulants, such as DOACs and acenocoumarol, were not significant in our analysis. Among these interactions, antiplatelet drugs such as clopidogrel and aspirin exhibited the highest prevalence. Omeprazole also emerged as a frequently interacting drug, albeit with mild severity. Our analysis revealed a strong association between clinical bleeding and abnormal coagulation results, as well as significant correlations between EGb drug interactions, bleeding risk, and abnormal coagulation. Notably, the severity of EGb drug interactions did not correlate with bleeding risk but did influence abnormal coagulation test results. Specific medications, including clopidogrel, aspirin, celecoxib, loxoprofen, nifedipine, and omeprazole, were significantly correlated with bleeding risk and abnormal coagulation. Overall, our findings underscore the importance of considering EGb drug interactions in bleeding risk assessments and coagulation management, highlighting the need for further research in this area to optimize patient safety and outcomes.

## Supporting information

**S1 Data. Data_ddi.**
(XLSX)

## Acknowledgments

We would like to express our gratitude to all the physicians and nurses in the Outpatient Clinic and Laboratory Department of Hai Phong International Hospital for their assistance in patient recruitment and clinical laboratory testing for our study. Furthermore, we greatly appreciate the invaluable support provided by Hai Phong University of Medicine and Pharmacy for our Biomedical - Pharmaceutical Sciences Research Group in this endeavor.

## Author contributions

**Conceptualization:** Ngo Thi Quynh Mai, Nguyen Viet Hieu, Tran Thi Ngan, Nguyen Thi Thu Phuong.

**Data curation:** Ngo Thi Quynh Mai, Nguyen Viet Hieu, Tran Thi Ngan, Tran Van Anh, Pham Van Linh, Nguyen Thi Thu Phuong.

**Formal analysis:** Ngo Thi Quynh Mai, Tran Thi Ngan, Nguyen Thi Thu Phuong.

**Investigation:** Ngo Thi Quynh Mai, Nguyen Thi Thu Phuong.

**Methodology:** Ngo Thi Quynh Mai, Nguyen Viet Hieu, Tran Thi Ngan, Tran Van Anh, Pham Van Linh, Nguyen Thi Thu Phuong.

**Project administration:** Nguyen Thi Thu Phuong.

**Software:** Nguyen Viet Hieu, Pham Van Linh, Nguyen Thi Thu Phuong.

**Supervision:** Pham Van Linh, Nguyen Thi Thu Phuong.

**Validation:** Ngo Thi Quynh Mai, Nguyen Thi Thu Phuong.

**Visualization:** Nguyen Thi Thu Phuong.

**Writing – original draft:** Ngo Thi Quynh Mai, Tran Thi Ngan, Tran Van Anh, Pham Van Linh, Nguyen Thi Thu Phuong.

**Writing – review & editing:** Pham Van Linh, Nguyen Thi Thu Phuong.

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
