## [Decision Letter · Decision Letter 0]

12 Nov 2024

PONE-D-24-33374Impact of Ginkgo biloba drug interactions on bleeding risk and coagulation profiles: A comprehensive analysisPLOS ONE

Dear Dr. Thi Thu,

Thank you for submitting your manuscript to PLOS ONE. After careful consideration, we feel that it has merit but does not fully meet PLOS ONE’s publication criteria as it currently stands. Therefore, we invite you to submit a revised version of the manuscript that addresses the points raised during the review process.

**The authors did well in the conceptualization of the study. However, the manuscript was not properly written. The result and discussion sessions need a major revision. There was no Figure 1 in the result session. I have highlighted areas where revision should be made.**

We look forward to receiving your revised manuscript.

Kind regards,

Akingbolabo Daniel Ogunlakin, Phd

Academic Editor

PLOS ONE

**Journal Requirements:**

Reviewers' comments:

Reviewer's Responses to Questions

**Comments to the Author**

1. Is the manuscript technically sound, and do the data support the conclusions?

Reviewer #1: Yes

Reviewer #2: Yes

2. Has the statistical analysis been performed appropriately and rigorously? 

Reviewer #1: Yes

Reviewer #2: Yes

3. Have the authors made all data underlying the findings in their manuscript fully available?

Reviewer #1: Yes

Reviewer #2: Yes

4. Is the manuscript presented in an intelligible fashion and written in standard English?

Reviewer #1: Yes

Reviewer #2: No

5. Review Comments to the Author

**Reviewer #1:**  This article is very well written and I recommend that this article is accepted for publication. the only typographical error i saw in this write-up in on line 78 where we have the sentence 'elderly patient who was concurrently using aspirin and Egb' the Egb in this sentence should be changed to EGb

**Reviewer #2:**  The authors did well in the conceptualization of the study. However, the manuscript was not properly written.

The result and discussion sessions need a major revision. There was no figure 1 in the result session.

I have highlighted areas where revision should be made.

6. PLOS authors have the option to publish the peer review history of their article (what does this mean? ). If published, this will include your full peer review and any attached files.

**Do you want your identity to be public for this peer review?** For information about this choice, including consent withdrawal, please see our Privacy Policy .

Reviewer #1: No

Reviewer #2: No

---

## [Author Response · Author response to Decision Letter 1]

17 Nov 2024

Response to reviewers:

The authors did well in the conceptualization of the study. However, the manuscript was not properly written. The result and discussion sessions need a major revision. There was no Figure 1 in the result session. I have highlighted areas where revision should be made.

Thank you for your feedback regarding our manuscript. We appreciate your recognition of the conceptualization of the study.

We acknowledge your comments about the need for major revisions in the results and discussion sections; we have taken the necessary steps to enhance clarity and coherence throughout these areas.

Regarding Figure 1, we understand that it was submitted separately in accordance with the guidelines of PLOS ONE. However, we will resubmit Figure 1 along with the revised manuscript to ensure it is readily available for review and aligns with the manuscript content.

We believe that the revisions made have significantly improved the manuscript, and we appreciate your valuable guidance in this process. Thank you for your consideration.

Line 23-40:

1. start the abstract with a little introduction on Gingo biloba

2. there is no methodology in this abstract. What was the selection criteria?

kindly include statistical analysis that was used.

Thank you for your insightful feedback regarding our abstract. We have made the recommended changes to enhance its clarity and comprehensiveness.

1. We have added an introductory statement about Ginkgo biloba at the beginning of the abstract to provide context for the study. This introduction helps establish the significance of Ginkgo biloba in relation to the research conducted.

2. We acknowledge the lack of methodological details in the original abstract. In response to your comments, we have included the selection criteria used for participant inclusion in the study. Additionally, we have specified the statistical analysis methods employed to assess the data.

The following paragraph has been added to our abstract to address these points:

“Our analysis utilized data from patients admitted to Hai Phong International Hospital between January 2022 and December 2023. Inclusion criteria consisted of patients aged 18 years and above, those prescribed Ginkgo biloba extract alone or in combination with other medications, and the availability of complete medical records, including medication history, laboratory tests, and clinical outcomes”

3. Add keywords

The following keywaords has been added to our manuscript:

Ginkgo biloba; drug interactions; bleeding risk; anticoagulants; coagulation profiles

Line 47 -48: reference???

In response to your inquiry about the reference for lines 47-48, we have added the appropriate citation to support the statements made in that section.

Line 51: repetition: G. biloba leaf extract (EGb) is widely recognized as one of the most popular health supplements globally due to its benefits for mental focus. EGb 761, the standardized EGb, is commonly used in studies investigating its effects. Research has indicated that EGb influences several neurotransmitter pathways and brain structures, particularly those observed in animal studies

In response to your feedback, we have revised this section to eliminate redundant phrases.

Line 73: Even though EGb is commonly used, it has risks, especially in terms of its potential interactions with prescription medications. These interactions can lead to issues like bleeding disorders and abnormal coagulation profiles. Consequently, caution is advised when administering EGb to patients with bleeding disorders or those who are taking NSAIDs, antiplatelet drugs, or anticoagulants.

In response to your feedback, we have revised this section based on your suggestions

Line 81: Recently, several case reports have suggested a possible connection between hemorrhagic complications and the use of EGb preparations. In response, a trial was carried out to evaluate the effects of the Ginkgo biloba special extract, EGb 761, on hemostatic parameters. The study found that, out of the 29 coagulation and bleeding parameters examined, EGb 761 did not show any evidence of inhibiting blood coagulation or platelet aggregation.

In response to your feedback, we have revised this section based on your suggestions

Line 114: Study design

In response to your feedback, we have revised this section based on your suggestions as following:

Patients aged 18 years and older

Patients prescribed EGb either alone or alongside other prescription medications

Availability of complete medical records, including medication history, laboratory tests, and clinical outcomes.

This criterion was specifically applied to examine the impact of EGb interactions on bleeding occurrences or abnormal coagulation tests.

Line 176-194: kindly summarize. some of the information should be moved to the discussion session

Discussion was poorly written.

Kindly rewrite this section.

1. the result was not properly discussed. kindly discuss the result . move some of the information in the results section to this section

2. authors should check for grammar. use the right tenses.

Thank you for your constructive feedback regarding the discussion section of our manuscript. We appreciate your attention to detail and the suggestions for improvement.

In response to your comments, we have rewritten the discussion to provide a more thorough analysis of the results. We have moved relevant information from the results section to ensure that findings are properly contextualized and discussed in detail.

Additionally, we have carefully reviewed the discussion for grammatical accuracy and proper use of tenses, making necessary adjustments to enhance clarity and readability.

We believe that the revisions have significantly strengthened the discussion section, and we appreciate your guidance in helping us improve the overall quality of the manuscript. Thank you for your valuable input.

---

## [Decision Letter · Decision Letter 1]

15 Jan 2025

PONE-D-24-33374R1Impact of Ginkgo biloba drug interactions on bleeding risk and coagulation profiles: A comprehensive analysisPLOS ONE

Dear Dr. Thi Thu,

Thank you for submitting your manuscript to PLOS ONE. After careful consideration, we feel that it has merit but does not fully meet PLOS ONE’s publication criteria as it currently stands. Therefore, we invite you to submit a revised version of the manuscript that addresses the points raised during the review process.

The authors have not addressed the comments raised in the previous review. I'll advise them to go through the previous reviewed document and address the comments especially the results and discussion section. I have also highlighted some areas that needs to be revised.

We look forward to receiving your revised manuscript.

Kind regards,

Akingbolabo Daniel Ogunlakin, Phd

Academic Editor

PLOS ONE

Additional Editor Comments :

The authors have not addressed the comments raised in the previous review. I'll advise them to go through the previous reviewed document and address the comments especially the results and discussion section. I have also highlighted some areas that needs to be revised

Reviewers' comments:

Reviewer's Responses to Questions

**Comments to the Author**

1. If the authors have adequately addressed your comments raised in a previous round of review and you feel that this manuscript is now acceptable for publication, you may indicate that here to bypass the “Comments to the Author” section, enter your conflict of interest statement in the “Confidential to Editor” section, and submit your "Accept" recommendation.

Reviewer #2: (No Response)

Reviewer #3: (No Response)

2. Is the manuscript technically sound, and do the data support the conclusions?

Reviewer #2: Yes

Reviewer #3: Yes

3. Has the statistical analysis been performed appropriately and rigorously? 

Reviewer #2: Yes

Reviewer #3: N/A

4. Have the authors made all data underlying the findings in their manuscript fully available?

Reviewer #2: Yes

Reviewer #3: Yes

5. Is the manuscript presented in an intelligible fashion and written in standard English?

Reviewer #2: No

Reviewer #3: Yes

6. Review Comments to the Author

Reviewer #2: The authors have not addressed the comments raised in the previous review. I'll advise them to go through the previous reviewed document and address the comments especially the results and discussion section. I have also highlighted some areas that needs to be revised.

Reviewer #3: The authors responded to most of the comments.

authors did NOT respond satisfactory to the following main points:

“Discussion was poorly written. Kindly rewrite this section. 1. the result was not properly discussed. kindly discuss the result . move some of the information in the results section to this section 2. authors should check for grammar. use the right tenses.”

Results were not adequately discussed.

7. PLOS authors have the option to publish the peer review history of their article (what does this mean? ). If published, this will include your full peer review and any attached files.

**Do you want your identity to be public for this peer review?** For information about this choice, including consent withdrawal, please see our Privacy Policy .

Reviewer #2: No

Reviewer #3: **Yes: ** Amel Elbasyouni

---

## [Author Response · Author response to Decision Letter 2]

24 Jan 2025

Response to Reviewers

Manuscript ID: PONE-D-24-33374R1

Title: Impact of Ginkgo biloba drug interactions on bleeding risk and coagulation profiles: A comprehensive analysis

Dear Dr. Ogunlakin and Reviewers,

We would like to thank you for the opportunity to revise our manuscript and for the constructive feedback provided by the reviewers. We have carefully considered all the comments and have revised the manuscript accordingly. Below, we provide a detailed response to each comment, including how we have addressed them in the revised manuscript.

Response to Reviewer #2

Comment 1: The authors have not addressed the comments raised in the previous review. I'll advise them to go through the previous reviewed document and address the comments, especially the results and discussion section. I have also highlighted some areas that need to be revised.

Response:

We regret that the previous revision did not fully address the concerns raised. In this revision, we have comprehensively reviewed the previous feedback and revised the manuscript accordingly, with a particular focus on the Results and Discussion sections. The following improvements have been made:

• Results Section:

• Key findings have been clarified and presented in a more structured format to enhance readability and alignment with the Discussion section.

• Statistical findings and their clinical relevance have been highlighted to provide a clear understanding of their implications.

• Discussion Section:

• The section has been rewritten to provide a detailed interpretation of the results in the context of existing literature.

• Relevant information from the Results section has been relocated to the Discussion where appropriate to ensure logical flow and avoid redundancy.

• Grammar and Clarity:

• The manuscript has been thoroughly revised for grammatical accuracy, proper tense usage, and improved readability.

Response to Reviewer #3

Comment 1: Discussion was poorly written. Kindly rewrite this section. The result was not properly discussed. Kindly discuss the result. Move some of the information in the Results section to this section.

Response:

We have completely rewritten the Discussion section to address this concern. Specific revisions include:

• A detailed analysis of key findings, including the prevalence and severity of drug interactions, correlations with clinical bleeding and coagulation test abnormalities, and implications for clinical practice.

• Integration of relevant literature to contextualize the findings and highlight their significance.

• Relocation of interpretative insights from the Results section to the Discussion to enhance logical flow and depth of analysis.

Comment 2: Authors should check for grammar. Use the right tenses.

Response: The manuscript has been thoroughly reviewed and revised for grammar, tense consistency, and overall readability.

General revisions across the manuscript

1. Statistical Analysis: We have double-checked the statistical methodologies and included clarifications where needed to ensure rigor.

2. Figures and Tables: All figures and tables have been reviewed for clarity and alignment with the text. We used the PACE digital diagnostic tool to confirm compliance with PLOS requirements.

3. Data Availability: We have reconfirmed the data availability statement and ensured that all data are accessible as described.

We believe these revisions address the reviewers’ and editor’s concerns and significantly enhance the quality of our manuscript. The revised manuscript with tracked changes and the clean version have been uploaded as separate files.

We appreciate your consideration and look forward to your feedback.

Sincerely,

---

## [Decision Letter · Decision Letter 2]

12 Mar 2025

Impact of Ginkgo biloba drug interactions on bleeding risk and coagulation profiles: A comprehensive analysis

PONE-D-24-33374R2

Dear Dr. Phuong Nguyen Thi Thu,

We’re pleased to inform you that your manuscript has been judged scientifically suitable for publication and will be formally accepted for publication once it meets all outstanding technical requirements.

Kind regards,

Akingbolabo Daniel Ogunlakin, Phd

Academic Editor

PLOS ONE

Additional Editor Comments (optional):

Reviewers' comments:

Reviewer's Responses to Questions

**Comments to the Author**

1. If the authors have adequately addressed your comments raised in a previous round of review and you feel that this manuscript is now acceptable for publication, you may indicate that here to bypass the “Comments to the Author” section, enter your conflict of interest statement in the “Confidential to Editor” section, and submit your "Accept" recommendation.

Reviewer #1: All comments have been addressed

Reviewer #4: (No Response)

2. Is the manuscript technically sound, and do the data support the conclusions?

Reviewer #1: Yes

Reviewer #4: Yes

3. Has the statistical analysis been performed appropriately and rigorously? 

Reviewer #1: Yes

Reviewer #4: Yes

4. Have the authors made all data underlying the findings in their manuscript fully available?

Reviewer #1: (No Response)

Reviewer #4: Yes

5. Is the manuscript presented in an intelligible fashion and written in standard English?

Reviewer #1: Yes

Reviewer #4: Yes

6. Review Comments to the Author

Reviewer #1: The authors have addressed all the reviewers comments, the topic is better phrased and the manuscript is better written.

Reviewer #4: (No Response)

7. PLOS authors have the option to publish the peer review history of their article (what does this mean? ). If published, this will include your full peer review and any attached files.

**Do you want your identity to be public for this peer review?** For information about this choice, including consent withdrawal, please see our Privacy Policy .

Reviewer #1: No

Reviewer #4: No

---

## [Editor Report · Acceptance letter]

PONE-D-24-33374R2

PLOS ONE

Dear Dr. Thu Phuong,

I'm pleased to inform you that your manuscript has been deemed suitable for publication in PLOS ONE. Congratulations! Your manuscript is now being handed over to our production team.

Kind regards,

on behalf of

Dr. Akingbolabo Daniel Ogunlakin

Academic Editor

PLOS ONE